# Feasibility and Effectiveness of Speech Intervention Implemented with a Virtual Reality System in Children with Developmental Language Disorders: A Pilot Randomized Control Trial

**DOI:** 10.3390/children10081336

**Published:** 2023-08-01

**Authors:** Irene Cappadona, Augusto Ielo, Margherita La Fauci, Maria Tresoldi, Carmela Settimo, Maria Cristina De Cola, Rosalia Muratore, Carmela De Domenico, Marcella Di Cara, Francesco Corallo, Emanuela Tripodi, Caterina Impallomeni, Angelo Quartarone, Francesca Cucinotta

**Affiliations:** IRCCS Centro Neurolesi Bonino Pulejo, 98124 Messina, Italy; irene.cappadona@irccsme.it (I.C.); augusto.ielo@irccsme.it (A.I.); margherita.lafauci@irccsme.it (M.L.F.); carmela.settimo@irccsme.it (C.S.); mariacristina.decola@irccsme.it (M.C.D.C.); rosalia.muratore@irccsme.it (R.M.); carmela.dedomenico@irccsme.it (C.D.D.); marcella.dicara@irccsme.it (M.D.C.); francesco.corallo@irccsme.it (F.C.); emanuela.tripodi@irccsme.it (E.T.); caterina.impallomeni@irccsme.it (C.I.); angelo.quatorone@irccsme.it (A.Q.); francesca.cucinotta@irccsme.it (F.C.)

**Keywords:** rehabilitation, virtual reality, speech intervention, language development, developmental language disorder, specific language impairment, randomized control trial, children

## Abstract

Language disorders are characterized by impairments in verbal expression/understanding, including difficulties with one or more language components. The Virtual Reality Rehabilitation System (VRRS) is a bioelectromedical device equipped with exercise sections aimed at improving cognitive and language deficits. It also increases patient motivation and engagement. The aim of our study was to test the feasibility and efficacy of VRRS intervention to improve speech therapy treatment for children with speech disorders. Thirty-two patients were enrolled in this study and randomly assigned to the experimental (EG) or control group (CG). The CG underwent conventional speech therapy, while EG underwent VRRS-implemented speech therapy. Both groups were evaluated before (T0) and after (T1) the intervention using the Language Assessment Test. The results showed improvements in both groups. However, the EG group showed greater improvement in various areas, including comprehension of total words, repetition, naming of body parts, naming of everyday objects, total naming, morphosyntactic accuracy, sentence construction, average length of utterance, and spontaneous word production. This study demonstrated that VRRS can be a valuable tool for implementing effective speech rehabilitation. Further studies are needed, as the use of VRRS is still in its early stages, requiring larger samples sizes and long-term follow-up.

## 1. Introduction

Developmental language disorder (DLD) is a neurodevelopmental condition characterized by persistent difficulty in the acquisition and use of language in all its modalities (spoken, written, sign language, or other). Children may present deficits in both comprehension and production, with a reduced number of words known and a persistent difficulty in their correct usage, limited ability to connect sentences to explain or describe a topic or series of events, or difficulty sustaining a conversation [1].

According to the literature, DLD is one of the most common disorders, and affects about 3–7% of preschoolers [2,3,4]. Language abilities in these children appear significantly below what is expected from subjects of the same age: children’s expressive language should be 50% intelligible by age 36 months and about 75% intelligible by age 48 months [5]. The onset of symptoms occurs in the early developmental period: sufficient receptive language seems to be positively correlated with better expressive language outcomes, such as toddlerhood vocabulary size [6]. Conversely, about 15–16% of toddlers with delays in both expression and comprehension of language presented a high risk of developing a persistent DLD [7]. DLD is described as a heterogeneous disorder because it may impair different aspects of language; often phonologic development is involved, and semantics or syntax difficulties are usually present [8]. Moreover, it is often associated with other neurodevelopmental disorders such as attention deficit hyperactivity disorder, specific learning disorder, autism spectrum disorder, and coordination development disorder [9,10,11]. These dysfunctions can lead to long-term disadvantages for the child, such as isolation, regression, and poor school performance [12]. Compared to other students, children and adolescents with a previous language disorder had a higher risk of reading or writing difficulties and are three times more likely to experience clinical levels of anxiety [13,14]. Overall, the outcome of an untreated DLD can result in functional limitations in effective communication, impaired peer interaction, reduction in academic achievement, and poor job retention [15]. To prevent comorbidity and improve outcome, specialist intervention is essential [12,16]. The best available evidence confirms the positive effect of speech and language therapy for children with DLD [17,18]. Speech therapy often involves exercises that can be boring, so it is necessary to motivate children to practice them through new techniques and innovative tools [19].

Over the past decade, virtual reality (VR) has played an increasing role in the treatment of several neurological disorders [20,21]. VR is a new methodology that involves the use of computer technologies that create various artificial environments (in 2D or 3D) similar to real ones that the patient will have to interact with through sight, sound, and touch [22,23,24]. The central nervous system, thanks to virtual scenarios, receives more sensory feedback (auditory, visual, tactile), which can create changes in synaptic plasticity and reinforce learning [25]. In this context, among the various tools involved in using virtual reality, the Virtual Reality Rehabilitation System (VRRS) developed by Khymeia (Padua, Italy) is a promising tool for rehabilitation [22,26,27]. It is a medical device with cognitive and linguistic modules that aim to improve the cognitive and linguistic abilities of patients with neurological disorders [22]. The application of VRRS could be beneficial for children with DLD as it increases motivation, reduces anxiety, and causes less attractive activities to be perceived as play. These latter elements are prerequisites for more effective recovery [28]. In this field, few studies have reported the usefulness of VR in communication disorders, mainly using it to mediate caregiver–child interaction [29] or to improve communication ability in patients with autism spectrum disorder. However, several clinical trials have been published that have shown the benefits of virtual reality in communication disorders, with results such as improvements in fluency disorders, pragmatic abilities, and interaction areas [30]. To our knowledge, this is the first prospective pilot study to verify the feasibility and clinical effectiveness of speech therapy implemented by the VRRS device in children with a developmental language disorder.

The purpose of this study is to test the feasibility and clinical effectiveness of speech therapy with the VRRS device in children with speech disorders, comparing the results with conventional treatment.

## 2. Materials and Methods

### 2.1. Population and Study Design

This pilot randomized control trial was developed in order to evaluate the feasibility and compare the efficacy of speech intervention implemented with a virtual reality system with treatment as usual (TAU) delivered to children affected by DLD. This study has been reviewed and approved by the Local Ethics Committee (approval number n. 15/2019) and completed in accordance with the CONSORT guidelines [31] (Figure 1). Written informed consent was obtained from both parents, or from a legal representative of the patient. Children were recruited at the IRCCS Bonino Pulejo Neurolesi Center in Messina, Italy, between February 2021 and December 2022. Recruited participants had a primary diagnosis of DLD as defined by the DSM-5 which was not attributable to sensory impairment or another medical or neurological condition. Other eligibility requirements were: age between 36 months and 84 months and a written informed consent provided by parents or guardians. Patients were excluded if they were receiving other behavioral therapy or medical treatments, and if they were diagnosed with other significant medical conditions such as epilepsy, visual and auditory sensory deficits, traumatic brain injury, or other significant genetic disorders. Patients included were randomly assigned to either the experimental group (EG) or the control group (CG) using a computer-generated list of random numbers. 

### 2.2. Neuropsychological Assessment

All patients underwent a complete and exhaustive assessment performed by expert clinicians. Diagnoses were made by a medical team comprised of pediatric neuropsychiatrists, psychologists, and speech therapists based on direct observation, a play session, a medical examination, a neuropsychological evaluation, and an interview with the parents. Children were assessed at baseline (T0) and at the end of the protocol (T1) with the Test of Language Assessment (TVL) [32] by an independent evaluator, blinded to treatment conditions. The TVL is an Italian standardized test that investigates language development in preschool children [33,34,35]. It is divided into four components: (1) receptive language, assessed through images representing body parts, common objects, colors, actions, objects, and spatiotemporal concepts; (2) sentence repetition ability, investigated by asking for repetition of 15 sentences of increasing length; (3) naming ability, examined by using pictures representing body parts and common objects and asking the child to name them; (4) spontaneous language production, assessed by asking the child to describe pictures representing actions, scenes, stories, and to narrate actions in sequence [36,37]. In addition, intellectual quotient (IQ) was assessed using a cognitive test (Leiter-3 [38], Wippsi-IV [39], Wisc-IV [40]). The tests administered were selected by child neuropsychiatrists based on the clinical characteristics of each patient. 

### 2.3. Intervention 

Each child participating in both groups received two 1-h sessions per week of treatment for a period of six months. The CG underwent conventional speech therapy, while the EG underwent speech therapy implemented by VRRS sessions. This virtual reality device is recognized as one of the most advanced, comprehensive, and clinically proven systems for rehabilitation [41,42]. It is a technological innovation tool to provide assistance with motor [43], cognitive, and linguistic performance [44,45]. VRRS allows the child to simulate daily activities in a non-immersive virtual environment, adapting task parameters according to the patient’s performance. This increases engagement and avoids boredom and frustration via a more sophisticated and holistic approach [42,46]. In our study, the speech therapy with the VRRS was planned by the speech therapist based on the neuropsychological assessment performed at T0. The tasks were the same for all children, but the difficulty and duration varied according to the needs and goals to be achieved. The exercises were 2D and the patient interacted with the scenarios through the touch screen. Table 1 shows the exercises proposed by speech therapists broken down by individual areas of focus. 

### 2.4. Statistical Analysis

Data were analyzed using R version 4.2.3, considering a *p* < 0.05 as statistically significant. A nonparametric analysis was performed. A Wilcoxon signed-rank test was used to compare scores between baseline and follow-up.

## 3. Results

A total of 45 children with DLD were screened for eligibility. Thirty-two children (mean age ± SD: 4.8 ± 1.1 years; M:F = 4.3:1) met the inclusion criteria and were randomly assigned to either the experimental group (EG: *n* = 16) or the control group (CG: *n* = 16). A more detailed description of the two groups is given in Table 2.

All patients involved in the study predominantly manifested DLD. Comorbid neurodevelopmental disorders were attention deficit hyperactivity disorder (n.6, 18.7%) and developmental coordination disorder (n.2, 6.3%). 

The feasibility appears to be well supported, as all patients accepted the procedures, attended regularly, and actively participated in the therapy, indicating that the DLD children were able to tolerate the virtual reality and the protocol. The retention rate of patients completing six months of treatment was 100%. Speech therapy led to improvement in both groups. In the EG group there were greater improvements in: comprehension—total words (*p* = 0.011), naming body parts (*p* = 0.033), naming everyday objects (*p* = 0.011), total naming (*p* = 0.029), morphosyntactic accuracy (*p* = 0.012), sentence construction (*p* = 0.030), average utterance length (*p* = 0.013), and spontaneous word production (*p* = 0.031). Significant improvements were not registered in comprehension—total sentences, construction—style, and construction—spontaneous production for the control group. For more details, see Table 3. 

## 4. Discussion

In our study, we compared traditional speech therapy treatment with a treatment aided by virtual technology by examining language at T0 and T1 through an assessment test of individual areas of language. The areas assessed were receptive language, sentence repetition ability, naming ability, and spontaneous language production. The treatment aimed to improve the same areas. Our study showed that speech therapy intervention with VRRS compared with conventional intervention made further improvements. In particular, an increase was observed in word comprehension, naming (of body parts, everyday objects, and totals), morphosyntactic accuracy, period construction, average utterance length, and spontaneous word production. 

It was not possible to compare all the areas of language investigated through TVL, as in the literature studies were not found that implement virtual reality and evaluate the improvement of language in children in all the components of TVL. This shortage is likely attributable to the lack of high-tech rehabilitation tools in hospitals due to high costs, accessibility issues, and lack of highly trained therapists [47]. Current studies are uneven and mainly concern adults and autism spectrum disorders. Recent studies of children with autism spectrum syndrome [48,49,50,51,52,53] show that VR made improvements in naming consonants [48], vowels [48], words, and sentences [49]. These results are in agreement with those of our study. In addition, VR improved progress in pragmatic skills by improving communication skills and social interaction [50,51,52,53]. Many authors have examined speech therapy treatment with VR in adults. It has been shown [54,55] that the authentic, safe, and controlled environments provided by VR can be useful for the assessment and treatment of stuttering [54] and that naming difficulties have been reduced with spontaneous reading and voice tests [55]. Again, our results are in line with these studies in terms of improved naming [54,55]. VR has also been applied to speech disorders following brain injury. In aphasia studies, general improvements were found in all areas analyzed, except writing [45] and learning, showing these improvements 2.3 times faster than in patients treated with conventional therapy [56]. 

Virtual reality is applicable in many areas of rehabilitation [50,57]. Such a tool is conducive to the treatment of communication disorders, particularly language disorders in children, as it promotes enjoyment, increases perception of self-efficacy, and facilitates engagement [58,59]. These elements allow for a longer duration of speech intervention and greater adherence to treatment [60]. The rehabilitative activities conducted with virtual reality in our research project resulted in improved speech. Some studies attribute the improvement predominantly to the simulation of reality [61,62]. Our research group believes that improvement is also related to the variety of stimuli selected and individualized for each child, mediation by therapists, and the experience of VR as a highly motivating playful situation [28,63,64]. 

Children also improve language skills through social and experiential games to which, however, they are increasingly less exposed, such as outdoor activities with animals and exposure to different stimuli [65]. Virtual reality, unlike experiential activities, encourages exposure to more stimuli and motivating activities, such as farm exercise, that reduce learning time and break down logistical barriers [66]. Digital stimuli induce greater frustration tolerance because even error is experienced as a game, and this facilitates improvement through repetition and trial-and-error procedures [66,67]. This technique can be useful because it allows therapists to present in a real-world-like setting [68] safe, controllable, and repeatable modes of intervention that enhance the development of children’s social, cognitive, and language skills [48]. Communication also becomes crucial, not only in relation to verbalization but also to socialization [69]. Another advantage of technological treatment is the reduction of rehabilitation time [70,71]. 

Overall, these results are in line with several studies conducted on DLD using VR, although our study is the first to provide a detailed picture of all the components of language that can improve in children through a speech intervention associated with VR investigated with TVL.

## 5. Conclusions

In conclusion, this study demonstrates that VRRS can be a valuable innovative tool for implementing feasible and effective speech rehabilitation, as it allowed greater recovery of abilities in different areas of language than usual treatment alone, thus promoting psychological well-being.

The VRRS is equipped with a large screen and contains many virtual exercises divided into specific categories in order to perform speech disorder training tailored to the specific needs of children.

It can therefore be considered a useful complementary treatment to reduce symptoms by providing fun and actively engaging children, increasing motivation and adherence to treatment. This is why VRRS should be widely implemented in clinical practice.

This is a preliminary study, as we are conducting further research to highlight what we found in our sample. However, our study has limitations in that it examines a very small sample with heterogeneous issues. The children also had possible comorbidities with other disorders. These factors might make the reader uncertain with respect to the methodology, however, we have positive results regarding the rapid recovery of language skills. The innovative element of our project concerns the rehabilitation of language components through high-tech tools. The technological aspects are appealing because they are immediate, allow for the breaking down of logistical barriers, and promote learning motivation through sounds, colors, and setting. To improve clinical practice in the future, more studies are needed as the use of VRRS is still in its early stages, so larger samples and long-term follow-ups are essential. 

## Figures and Tables

**Figure 1 children-10-01336-f001:**
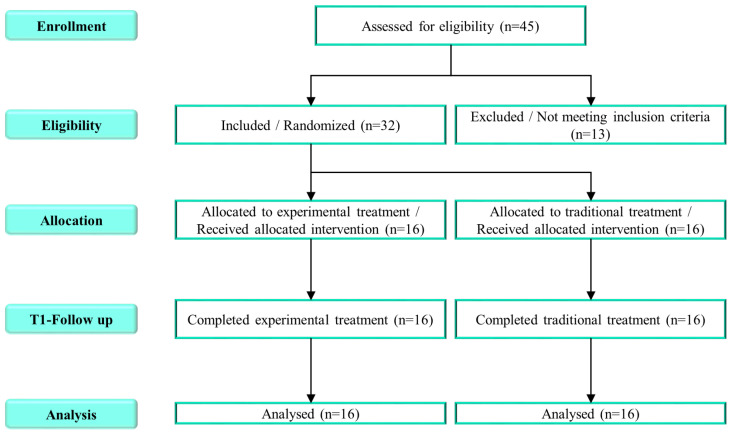
The CONSORT flowchart with detailed information on participants in the study.

**Table 1 children-10-01336-t001:** Description of the main exercises used during session with VRRS and language domain involved in each activity.

Exercise	Game	Language Domain
Phrase Selection	Select the phrase representing the video shown.	Comprehension
Identify Action	Write the action performed in the video.	Comprehension
Semantic Recognition	Recognize the object among the images depicted	Comprehension
Image Recognition Acoustic Reference	Recognize the object among the images represented with acoustic reference.	Comprehension
Similar Word Recognition	Recognize the object among the similar words represented.	Comprehension
Separate by Color	Click on objects in the same category to divide them.	Comprehension
Separates by Shape	Drag and drop objects into the correct category.	Comprehension
Separate by Semantic Group	Drag and drop objects into the correct category.	Comprehension
Spatial Perceptual Orientation Front/Back	Drag the objects into the correct category by means of the verbal prose in front and behind.	Comprehension
Recognize One Object Among Many	Select from the list the identical item to be indicated.	Comprehension
Select Phrase	Repetition of sentences with picture help.	Repetition
Tachistoscope Pictures	Repeat the images depicted in a period of a few seconds.	Repetition
Tachistoscope Words	Repeat the words represented in a period of a few seconds.	Repetition
Farm	Repeat and select animals equal to those represented in the box.	Repetition
Advanced Point Connection	Join the dots ordered numerically and alphabetically.	Denomination
Tachistoscope Images	Name the images depicted in a period of a few seconds.	Denomination
Tachistoscope Words	Name the words represented in a few seconds.	Denomination
Reorder the Syllable Count	Rearrange the syllables to reconstruct the word.	Verbal Production
Select the Syllable Count	Select the first syllable belonging to the indicated image.	Verbal Production
Select Images	Select from multiple images represented.	Verbal Production
Subdivide Images	Divide between several images represented.	Verbal Production
Identify Action	Write down the action performed in the video.	Verbal Production
Identify Sentence	Write the completed sentence in the video.	Verbal Production

**Table 2 children-10-01336-t002:** Demographic data of the sample.

	EG		CG		Total Sample	
	N	Mean ± SD or %	N	Mean ± SD or %	N	Mean ± SD or %
**Participants**	16	50%	16	50%	32	
**Age (years)**		4.9 ± 1.3		4.7 ± 1.0		4.8 ± 1.1
**Gender**						
*Male *	13	81.3%	13	81.3%	26	81.3%
*Female *	3	18.7%	3	18.7%	6	18.7%
*M:F Ratio *	4.3:1		4.3:1		4.3:1	
**Comorbid Neurodevelopmental Disorders**						
*ADHD*	3	18.7%	3	18.7%	6	18.7%
*DCD*	1	6.3%	1	6.3%	2	6.3%

Legend: experimental group (EG); control group (CG). Mean ± standard deviation was used to describe continuous variables; proportions (numbers and percentages) were used to describe categorical variables. Attention deficit hyperactivity disorder (ADHD); developmental coordination disorder (DCD).

**Table 3 children-10-01336-t003:** Statistical comparisons of clinical scores between baseline (T0) and follow-up (T1), for both experimental and control groups.

Clinical Assessment	Experimental Group		*p*-Value	Control Group		*p*-Value
	T0	T1		T0	T1	
**COMP.WORDS**	3.5	9.0	**0.011**	5.0	5.5	**0.036**
	(2.0–7.0)	(3.0–10.0)		(1.5–7.5)	(3.8–10.0)	
**COMP.SENT**	2.0	7.0	**0.004**	3.0	5.0	0.059
	(3.0–10.0)	(2.0–10.0)		(0.0–5.3)	(3.0–7.0)	
**COMP.TOTAL**	3.5	9.0	**0.004**	4.5	5.5	**0.022**
	(2.0–5.5)	(3.0–10.0)		(1.5–5.3)	(3.0–10.0)	
**REPETITION**	2.0	4.5	**0.004**	2.5	3.5	**0.003**
	(0.0–10.0)	(2.8–7.0)		(0.0–4.0)	(2.0–5.3)	
**NAMING.BODY**	2.5	6.5	**0.033**	2.5	5.0	**0.009**
	(0.0–7.0)	(3.8–9.3)		(0.0–5.3)	(3.0–9.0)	
**NAMING.OBJ**	6.5	9.0	**0.011**	4.0	5.0	**0.009**
	(0.0–9.3)	(5.3–10.0)		(0.0–6.0)	(3.0–9.3)	
**NAMING.TOTAL**	3.0	6.5	**0.029**	3.5	5.0	**0.004**
	(0.0–7.0)	(2.8–10.0)		(0.0–5.0)	(3.0–9.3)	
**ACCURACY.PHONO**	0.0	3.5	**0.009**	1.0	2.5	**0.002**
	(0.0–3.3)	(2.0–6.0)		(0.0–3.0)	(2.0–6.3)	
**ACCURACY.MORPH**	2.0	6.5	**0.012**	3.0	4.5	**0.033**
	(0.0–5.5)	(3.8–10.0)		(0.0–7.5)	(2.0–10.0)	
**CONS.SENT**	0.0	5.0	**0.005**	2.5	6.0	**0.002**
	(0.0–4.3)	(2.0–9.3)		(0.0–4.5)	(2.80–10.0)	
**CONS.PERIOD**	0.0	2.5	**0.030**	3.0	6.0	**0.002**
	(0.0–3.3)	(2.0–6.8)		(0.0–6.0)	(2.8–10.0)	
**CONS.MLU**	1.0	5.0	**0.013**	2.5	4.0	**0.039**
	(0.0–4.5)	(2.0–9.3)		(0.0–5.0)	(1.5–9.3)	
**CONS.STYLE**	2.5	6.5	**0.006**	2.0	4.0	0.090
	(0.0–6.0)	(2.8–7.5)		(0.0–4.3)	(0.0–6.0)	
**CONS.SPONT**	2.0	3.5	**0.031**	3.0	4.5	0.071
	(1.5–4.3)	(2.0–6.3)		(0.0–4.0)	(2.0–5.0)	

Scores are in median (first–third quartile); significant differences between treatment effects are in bold. Legend: COMP.WORDS = comprehension: total words; COMP.SENT = comprehension: total sentences; COMP.TOTAL = comprehension: total; REPETITION = repetition; NAMING.BODY = naming of body parts; NAMING.OBJ = naming of objects; NAMING.TOTAL = naming: total; ACCURACY.PHONO = phonological accuracy; ACCURACY.MORPH = morphosyntactic accuracy; CONS.SENT = sentence construction; CONS.PERIOD = period construction; CONS.MLU = construction: mean length of utterance; CONS.STYLE = construction: style; CONS.SPONT = construction: spontaneous production.

## Data Availability

The data presented in this study are available on request from the corresponding author.

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
