# Peer review of "Feasibility and Effectiveness of Speech Intervention Implemented with a Virtual Reality System in Children with Developmental Language Disorders: A Pilot Randomized Control Trial"

_children, 2023, doi:10.3390/children10081336_

Round 1

Reviewer 1 Report

The introduction is very well done. Well argued and with impact and updated quotes.

The Figures must be sent with higher quality. The text on them looks blurry.

The results are very well presented. The tables are very enlightening and pertinent. The information is not repeated, and important ideas are highlighted in the text, and the entire data set is collected in the tables.

  The discussion is very interesting. Similar studies are used to compare the results obtained in this investigation with those of said studies.

The references are abundant, they are current, impactful and highly relevant.

The conclusions are very short. The authors are invited to write something else, so that the conclusions are more direct and accurate.

Author Response

Dear reviewer, thank you for taking the time to review this research paper. We appreciate your thoughtful feedback. Here are our replies to your comments, where needed.

Reviewer: The Figures must be sent with higher quality. The text on them looks blurry.

Author: Figure 1 have been replaced with a better quality version.

Reviewer: The conclusions are very short. The authors are invited to write something else, so that the conclusions are more direct and accurate.

Author: Thank you for your suggestion. Conclusions have been enriched.

Reviewer 2 Report

An original and randomized study about language difficulties and a treatment with virtual reality is presented. The study presents a good introduction, concrete and clear. Collect relevant and updated studies. The methodology is considered adequate, well implemented and correct to achieve the proposed objectives. The results are pertinent and the discussion and conclusions respond to the ideas initially exposed. The graphs and tables give an account of what has been done and offer a pertinent overview for the reader.

As a suggestion: Indeed, there are few studies that relate the variables that you indicate, therefore, it is interesting that you pose research questions and explicitly describe the objectives of your research in a specific section before the method.

Author Response

Dear reviewer, thank you for taking the time to review this research paper. We appreciate your thoughtful feedback. Here are our replies to your comments, where needed.

Reviewer: Indeed, there are few studies that relate the variables that you indicate, therefore, it is interesting that you pose research questions and explicitly describe the objectives of your research in a specific section before the method.

Author: Thank you for your suggestion. The study objective was added at the end of the introduction section.

Reviewer 3 Report

·         Line 114: Please clarify what a medical equip is or change the sentence to, “Diagnoses were made by a medical team comprised of pediatric neuropsychiatrists, psychologists, and speech therapists based on direct observation, a play session, a medical examination, a neuropsychological evaluation, and an interview with the parents.”

Figure 1:

·         The numbers in this figure are confusing. It starts with 45 participants screened, and says 13 declined to participate, but the next line shows them as “not meeting inclusion criteria.” The third row down is also numerically unclear.

Table 2:

The “Accessing additional therapies,” part of this table is unnecessary as it is already made clear that it is an exclusion criterion in the materials and methods section

Thank you for requesting our comments on your manuscript titled “Feasibility and effectiveness of speech intervention implemented with a virtual reality system in children with developmental language disorders: a pilot randomized control trial.”. It is a novel study, and would make a great contribution to the literature, however, it does need extensive edits prior to publication. 

Abstract:

·         Line 25: Take out the word “we,” as it is the only sentence in that tense. For example: “The EG group showed greater improvement […]”

·         Line 26: The dash between “comprehension” and “total words.,” should be changed to a comma.

·         Line 28: Needs “and” before “spontaneous word production.”

·         Line 31: Change to “sizes.”

Introduction:

·         Line 43: Change to “According to the literature, DLDs is one of the most common disorders, and affects about 3%-7% of preschoolers.

·         Line 44-46: Language abilities result substantially and quantifiably below what is expected for chronological age.” This sentence does not make sense, please clarify.

·         Line 47: Take out “a” before “sufficient receptive language.”

·         Line 48: Change to “outcomes.”

·         Line 49: Change to “toddlers,” and change to “delays.”

·         Line 53: Present should be one word.

·         Line 55: Add “and” before “coordination development disorder.”

·         Line 55-62: Change these sentences to “These dysfunctions can lead to long-term disadvantages for the child such as isolation, regression, and poor school performance. Compared to other students, children and adolescents with a previous language disorder had a higher risk of reading or writing difficulties and are 3 times more likely to experience clinical levels of anxiety; overall, the outcome of an untreated DLD can result in functional limitations in effective communication, impaired peer interaction, reduction in academic achievement, and poor job retention.”

·          Line 63: Change the word ameliorate.

·         Line 64: Change to “[…] confirms the positive effect of […].”

·         Line 78: Replace “because,” with “as.”

·         Line 97: Change this sentence to, “Recruited participants had a primary diagnosis of DLD as defined by the DSM-5 and not attributable to sensory impairment […].”

·         Line 101: Change “caregivers,” to “parents or guardians,” for consistency.

·         Line 104: Remove repeated words, “to either.”

·         Line 124: Change to “[…] naming ability, examined by using pictures representing body parts and common objects and asking the child to name them.”

·         Line 127: Change “was,” to “were.”

·         Line 133: Change “performed,” to “underwent.”

·         Line 176: Change to, “In the EG group there were greater improvements in: […].”

·         Line 180: Change to, “Significant improvements were not registered in comprehension – total sentences, construction – style and, construction – spontaneous production for the control group.”

·         Line 196: Capitalized T0 and T1 for consistency.

·         Line 204-205: This sentence is not very clear as there is little background on what a single component is in the rest of the article.

·         Line 208: Remove “In fact,”

Author Response

Dear reviewer, thank you for taking the time to review this research paper. We appreciate your thoughtful feedback. Here are our replies to your comments, where needed.

Reviewer: Line 114: Please clarify what a medical equip is or change the sentence to, “Diagnoses were made by a medical team comprised of pediatric neuropsychiatrists, psychologists, and speech therapists based on direct observation, a play session, a medical examination, a neuropsychological evaluation, and an interview with the parents.”

Authors: The mentioned sentence have been replaced. Thank you for your suggestion.

Reviewer: Figure 1: The numbers in this figure are confusing. It starts with 45 participants screened, and says 13 declined to participate, but the next line shows them as “not meeting inclusion criteria.” The third row down is also numerically unclear.

Author: Thank you for your feedback. Numbers in the figure have been updated.

Reviewer: Table 2: The “Accessing additional therapies,” part of this table is unnecessary as it is already made clear that it is an exclusion criterion in the materials and methods section

Author: Thank you for your feedback. "Accessing additional therapies" part of Table 2 have been removed.

Comment on the English language: All the suggestions have been applied. Thank you. Regarding the "Line 204-205: This sentence is not very clear as there is little background on what a single component is in the rest of the article." comment, we added some background on the TVL test inside the "2.2. Neuropsychological assessment" section. We are referring to the single components of the TVL test. The sentence has been rewritten to be more comprehensible.